# Antibiotic Use in Pediatric Care in Ghana: A Call to Action for Stewardship in This Population

**DOI:** 10.3390/antibiotics14080779

**Published:** 2025-08-01

**Authors:** Israel Abebrese Sefah, Dennis Komla Bosrotsi, Kwame Ohene Buabeng, Brian Godman, Varsha Bangalee

**Affiliations:** 1School of Pharmacy, University of Health and Allied Sciences, Ho PMB 31, Ghana; kobuabeng@uhas.edu.gh; 2Department of Epidemiology and Biostatics, Fred Binka School of Public Health, University of Health and Allied Sciences, Ho PMB 31, Ghana; dbosrotsi25pg@sph.uhas.edu.gh; 3Department of Public Health Pharmacy and Management, School of Pharmacy, Sefako Makgatho Health Sciences University, Ga-Rankuwa 0208, South Africa; 4Strathclyde Institute of Pharmacy and Biomedical Sciences, University of Strathclyde, Glasgow G4 0RE, UK; 5Antibiotic Policy Group, City St. George’s, University of London, London SW17 0RE, UK; 6Discipline of Pharmaceutical Sciences, School of Health Sciences, University of KwaZulu-Natal, Durban 4041, South Africa; bangalee@ukzn.ac.za

**Keywords:** Ghana, pediatric patients, point prevalence survey, antibiotics, AWaRe classification, guidelines, appropriateness, hospitals, ceftriaxone

## Abstract

**Background/Objectives**: Antibiotic use is common among hospitalized pediatric patients. However, inappropriate use, including excessive use of Watch antibiotics, can contribute to antimicrobial resistance, adverse events, and increased healthcare costs. Consequently, there is a need to continually assess their usage among this vulnerable population. This was the objective behind this study. **Methods**: The medical records of all pediatric patients (under 12 years) admitted and treated with antibiotics at a Ghanaian Teaching Hospital between January 2022 and March 2022 were extracted from the hospital’s electronic database. The prevalence and appropriateness of antibiotic use were based on antibiotic choices compared with current guidelines. Influencing factors were also assessed. **Results**: Of the 410 admitted patients, 319 (77.80%) received at least one antibiotic. The majority (68.65%; n = 219/319) were between 0 and 2 years, and males (54.55%; n = 174/319). Ceftriaxone was the most commonly prescribed antibiotic (20.69%; n = 66/319), and most of the systemic antibiotics used belonged to the WHO Access and Watch groups, including a combination of Access and Watch groups (42.90%; n = 136/319). Neonatal sepsis (24.14%; n = 77/319) and pneumonia (14.42%; n = 46/319) were the most common diagnoses treated with antibiotics. Antibiotic appropriateness was 42.32% (n = 135/319). Multivariate analysis revealed ceftriaxone prescriptions (aOR = 0.12; CI = 0.02–0.95; *p*-value = 0.044) and surgical prophylaxis (aOR = 0.07; CI = 0.01–0.42; *p*-value = 0.004) were associated with reduced antibiotic appropriateness, while a pneumonia diagnosis appreciably increased this (aOR = 15.38; CI = 3.30–71.62; *p*-value < 0.001). **Conclusions**: There was high and suboptimal usage of antibiotics among hospitalized pediatric patients in this leading hospital. Antibiotic appropriateness was influenced by antibiotic type, diagnosis, and surgical prophylaxis. Targeted interventions, including education, are needed to improve antibiotic utilization in this setting in Ghana and, subsequently, in ambulatory care.

## 1. Introduction

Antimicrobial resistance (AMR) is a growing global concern with increasing morbidity and mortality exacerbated by inappropriate antibiotic use [1,2,3,4]. As a result, AMR is increasingly seen as the next pandemic unless appreciable activities are undertaken to help reverse rising AMR rates [5]. This is particularly important among low- and middle-income countries (LMICs), including Sub-Saharan African countries, where the burden of AMR is greatest [6,7]. The economic consequences associated with AMR can also be considerable, with estimates that by 2050, AMR could reduce the gross domestic product per country by up to 3.8% [8,9].

The pediatric population is at greater risk of AMR than other populations; consequently, this age group should be appropriately managed with antibiotics to reduce future AMR-related morbidity and mortality [10]. Considerable rates of multi-drug-resistant organisms (MDROs) have been reported in Africa in this population, which, in some studies, was up to 66–90%, increasing AMR [11,12,13,14,15,16]. These observations may have resulted from the increased inappropriate prescribing of antibiotics of 50% or more among pediatric patients [16,17,18,19]. It was estimated in 2022 that more than 752,000 children in Southeast Asia and 659,000 children across Africa died of AMR-associated complications [20,21], with over 1.5 million more deaths in other World Health Organization (WHO) regions from AMR-associated complications [20]. Many of these deaths were linked to increased use of Watch and Reserve antibiotics, with their higher resistance potential [20,21,22], with the use of Watch and Reserve antibiotics increasing by 126% and 125%, respectively, in this population in Africa between 2019 and 2021 [21]. The is a concern with the pediatric population, typically exposed to antibiotics more than any other form of medication due to their susceptibility to infections [16], subject to rising antibiotic consumption in recent years [23]. There is, though, a lack of data on AMR-associated mortality among pediatric patients in Ghana; however, there is a high and growing prevalence of AMR in Ghana, causing concern [24].

There have been several global, regional, and national initiatives in recent years to address the issues with rising AMR rates. These include the instigation of the WHO Global Action Plan (GAP) to reduce AMR in 2015 [25], subsequently translated into national action plans (NAPs) [26,27,28]. This includes Ghana; however, there have been concerns with the implementation of NAPs across Africa due to resource and personnel issues [29,30,31]. Suggested NAP activities include greater monitoring of current antimicrobial utilization rates across sectors, as well as the instigation of antimicrobial stewardship (AMS) activities, which include the implementation of antimicrobial stewardship programs (ASPs), with their known impact on improving future antibiotic utilization [32,33,34].

Other global initiatives include the development of the WHO essential medicines list, classifying antibiotics into three categories, namely, Access, Watch, and Reserve (AWaRe) antibiotics, based on their resistance potential [22,35]. Under this system, antibiotics from the Access group, with their narrow spectrum of activity, are encouraged where antibiotic prescribing or dispensing is pertinent, due to their generally low resistance potential [22,35]. Antibiotics from the Watch group, which include azithromycin as well as third-generation cephalosporins such as ceftriaxone, should ideally be prescribed only in critical conditions due to their broader-spectrum activity and greater resistance potential. Antibiotics from the Reserve group, including colistin, should only be prescribed in multi-drug-resistant cases as they are last-resort antibiotics [22,35]. This helps to explain why there is concern with the substantial increase in the use of Watch and Reserve antibiotics among children in Africa between 2019 and 2021 [21].

In 2022, the WHO launched the WHO AWaRe book, giving guidance on the management of 35 infections typically seen across sectors, with guidance subsequently moderated depending on local resistance patterns and updates [36,37,38,39]. It is increasingly likely that indicators to improve future antibiotic prescribing will be based on reducing the utilization of Watch and Reserve antibiotics [40]. Indicators are also likely to include encouraging greater adherence to well-accepted and robust guidelines now that the WHO AWaRe book guidelines are available, alongside the increasing use of the WHO AWaRe system to track antibiotic utilization patterns [36,38,39,40,41,42]. Such activities will be encouraged by the recent United Nations General Assembly (UN GA) goal to reduce AMR, which includes a new target of at least 70% of antibiotic use across sectors being from the Access group [43]. Low- and middle-income countries (LMICs), including Sub-Saharan African countries, will be critical in this respect to reverse increasing consumption of Watch and Reserve antibiotics, including among children, and their subsequent impact on AMR [21,44,45]. Published studies in LMICs have shown that increased audit and education surrounding the AWaRe classification and guidance can decrease inappropriate prescribing of Watch and Reserve antibiotics, including in Ghana [46,47].

We are aware that there have been concerns with inappropriate prescribing of antibiotics across the sectors in Ghana, including among children and during the COVID-19 pandemic, exacerbated by a lack of culture and sensitivity testing (CST) and compliance with treatment guidelines [48,49,50,51,52,53,54,55,56,57,58,59]. The lack of routine CST activities among hospitals in Ghana is not helped by high patient co-payments for testing [49,51,58,60]. Concerns with guideline compliance among hospitals in Ghana include the Ho Teaching Hospital (HTH), where a point prevalence survey (PPS) undertaken in 2020 showed a high prevalence of antibiotic use among children, exacerbated by a poor guideline compliance rate of 33% [51,61,62]. Infections in Ghana’s pediatric population are an appreciable concern, with pneumonia, diarrhea, and malaria being the leading causes of morbidity and mortality in this population [53,61,62]. Limited access to medical care, especially in rural areas, alongside malnutrition, exacerbates the situation [63]. However, we are also aware of ongoing activities to improve future antibiotic prescribing in Ghana. This includes the implementation of the AMR NAP coupled with the establishment of a national AMR platform to ensure coordinated interventions, enhancing antibiotic and microbiological data surveillance, along with other ASP activities, which should all help address some of the current concerns [15,47,64,65].

Consequently, given previous findings and activities, there is a need to assess the current prevalence and appropriateness of antibiotic use initially among hospitalized pediatric patients presenting with common infectious diseases in Ghana, alongside key factors influencing treatment choices. Given this objective, this paper aims to ascertain current antibiotic utilization patterns among hospitalized children in a tertiary hospital in Ghana, their appropriateness, and key factors influencing antibiotic choices. The findings can be used to suggest targeted interventions where pertinent to improve future antibiotic prescribing among this vulnerable hospitalized population. This is seen as imperative, with pediatricians in tertiary hospitals playing a key role in educating future prescribers in primary care. If there are concerns with the antibiotic-prescribing habits and knowledge regarding AMS among hospital pediatricians [66], these concerns will be magnified once trained physicians and other healthcare professionals start treating children in ambulatory care in Ghana.

## 2. Results

The electronic medical records of pediatric patients (children under 12 years) admitted and treated with antibiotics at the Ho Teaching Hospital over a three-month duration (January 2022 to March 2022) were collected and analyzed.

### 2.1. Patient Characteristics and Their Association with Guideline Compliance

The total number of pediatric patients hospitalized within the three-month study period was 410. Out of this, 319 patients were prescribed an antibiotic, giving a prevalence rate of 77.80%. With a mean age of 2.39 ± 2.99 years, the majority (68.65%; n = 219/319) of patients were between the ages of 0 and 2 years, followed by those between the ages of 3 and 5 years (16.61%; n = 53/319). Most of the children were male (54.55%; n = 174/319) and lived in the rural part of the Ho Municipality (46.86%; n = 149/318) (Table 1).

The mean duration of admission was 9.14 ± 10.73 days, while that of antibiotic therapy was 3.03 ± 2.43 days. Most (96.55%; n = 308/319) of the patients’ treatment was based on biomarker findings, with white blood cell count (WBC) (41.56%; n = 128/308) being the most commonly used biomarker, followed by a combination of WBC with C-reactive protein (CRP) (26.30%; n = 81/308).

The majority (56.74%; n = 181/319) of the pediatric patients were prescribed two antibiotics, followed by those who were given a single antibiotic (37.62%; N = 120/319). The commonest antibiotics prescribed were ceftriaxone (20.69%; n = 66/319), followed by a combination of cefotaxime + flucloxacillin (15.99%; n = 51/319) and ampicillin + gentamicin (15.05%; n = 48/319). Most of the systemic antibiotics prescribed were a combination of the WHO Access and Watch groups (42.90%; n = 136/319), followed by those from the Watch group alone (29.97%; n = 95/319). Most antibiotics were administered intravenously (82.76%; n = 264/319), followed by a combination of intravenous and oral (IV + PO) routes (8.46%; n = 27/319).

Neonatal sepsis (24.14%; n = 77/319), followed by pneumonia (14.42%; n = 46/319) and neonatal jaundice arising from neonatal sepsis (5.64%; 18/319), were the most common infectious diseases managed with antibiotics among these neonates and children.

The level of appropriateness of antibiotic prescriptions for these patients based on guideline compliance was 42.32% (n = 135/319). The greatest compliance among the various infectious diseases diagnosed was seen in the management of childhood pneumonia (84.78% of patients), with the least compliance among reasonable patient numbers seen in those neonates and children undergoing surgery to prevent wound infections (Table 1). Overall, the level of appropriateness, i.e., guideline compliance versus non-compliance, was associated with the type of antibiotic prescribed (*p*-value < 0.001), the class of antibiotic prescribed (*p*-value < 0.001), the WHO AWaRe classification of the antibiotic (*p*-value < 0.001), the route of administration of the antibiotic (*p*-value = 0.009), and the diagnosis (*p*-value < 0.001) (Table 1).

### 2.2. Predictors of Level of Appropriateness Based on Treatment Guideline Compliance

The level of appropriateness of antibiotic prescribing based on the choice of antibiotic was reduced by 88% when ceftriaxone (aOR = 0.12; CI = 0.02–0.95; *p*-value = 0.044) was prescribed compared with that of amoxicillin with clavulanic acid.

The appropriateness level was increased by approximately 15 times when pneumonia (aOR = 15.38; CI = 3.30–71.62; *p*-value < 0.001) was diagnosed compared with a neonatal sepsis diagnosis; however, appropriateness levels were reduced by 93% when antibiotics were given for surgical prophylaxis (aOR = 0.07; CI = 0.01–0.42; *p*-value = 0.004) compared with when neonatal sepsis was diagnosed (Table 2). Overall, this showed a statistically significant association and guideline compliance based on antibiotic choice.

## 3. Discussion

This study found a high prevalence of antibiotic use among hospitalized pediatric patients in a leading tertiary hospital in the Volta Region, similar to previous studies in Ghana and other LMICs [51,52,61,67,68,69,70,71,72,73,74]. However, other studies in LMICs have demonstrated a lower prevalence of antibiotic use among this population [75,76]. The principal concern in HTH was that more than 50% of the pediatric population were given a combination of at least two broad-spectrum antibiotics, which included antibiotics from the Watch group. This has implications for increasing the risk of AMR and must be the subject of future ASPs in this hospital as well as similar hospitals across Ghana [38,45].

We are aware of recent activities in Ghanaian hospitals to increase the instigation of ASPs given concerns with current hospital antibiotic utilization patterns. This has been helped by partnerships with UK institutions under the umbrella of the Commonwealth Partnerships for promoting ASPs alongside other activities [15,34,47,60,64,77]. However, we are also aware that the implementation of ASPs has been challenging across LMICs, including Ghana, and among pediatricians, due to factors such as health system challenges, which include inadequate diagnostic infrastructure, inadequate capacity building, poor supply chains, and unregulated antibiotic access [66,78]. This, though, is now changing across Africa, including Ghana, where studies have shown promising results from ASP interventions [34,47,70,71,72,73,74,75,76,77,78,79,80,81,82]. This is important to attain UN GA targets of increased use of Access antibiotics, with their lower risk of causing AMR, especially given the increasing use of Watch antibiotics among children in Africa in recent years [21,22,32,43,65].

The low level of adherence to current guidelines in our study, at only 43.2% of antibiotic prescriptions, is of utmost concern, alongside the prescribing of irrational antibiotic combinations. Poor adherence may have stemmed from limited awareness or understanding of current guidelines, and the AWaRe classification of antibiotics, similar to other studies in Ghana and other LMICs [49,51,74,83,84]. Additional obstacles may have included inadequate training, lack of knowledge regarding ASPs, access to appropriate resources, and high patient loads that pressure quick decision making [60,66,84]. These barriers, though, were not assessed in this study. However, they will be explored further in future studies, considering their beneficial impact on subsequent antibiotic use with increasing adherence to robust guidelines. Whilst the irrational use of antibiotics has been observed in other countries and settings among this population [74,85,86,87,88,89], this needs to be urgently addressed in HTH and across Ghana to achieve NAP and UN GA AMR goals [30,43].

Overall, the appropriateness of antibiotic prescribing in our study was associated with the type of antibiotics, the WHO AWaRe group, the route of administration, and the type of infectious disease diagnosed (Table 1). Among the clinical characteristics assessed in this study, increased appropriateness was predicted by a diagnosis of pneumonia and reduced among those patients given antibiotics for surgical prophylaxis (Table 2). This is partly encouraging, with pneumonia being one of the leading causes of admission among children in this hospital and elsewhere across LMICs [50,61,74,88,90]. Prolonged use of antibiotics for surgical prophylaxis, though, is a concern, increasing adverse reactions, AMR, and healthcare costs [6,91].

Another key concern was the appreciable prescribing of ceftriaxone, either alone or in combination with the current Ghanaian STG, prioritizing the prescribing of a penicillin combination, namely, amoxicillin plus clavulanic acid or ampicillin with gentamicin, over the use of third-generation cephalosporins for the management of commonly diagnosed pediatric conditions [92]. These include respiratory tract infections, neonatal sepsis, and antibiotics for surgical prophylaxis. In addition, third-generation cephalosporins are in the WHO ‘Watch’ group, with their increased risk of AMR development [22,35]. Third-generation cephalosporins are especially noted for increased selective pressure that precipitates genetic mutation for AMR genes, leading to treatment failures and, consequently, increased mortality [93,94]. As a result, their prescribing should be avoided where possible. The recent publication of the WHO AWaRe book, giving guidance for the optimal management of 35 infectious diseases across sectors, should help in this regard, enhanced by recent additions as well as quality indicators based on AWaRe percentage rates and adherence to the WHO AWaRe book guidance [36,37,38,39,40].

Whilst the studied children were exposed to antibiotics for approximately three days, the duration of hospital stay was approximately nine days. This suggests that despite a relatively short duration of antibiotic treatment, hospitalization was prolonged, potentially increasing their risk of healthcare-associated infections and healthcare costs [95,96]. This highlights the urgent need for optimized care pathways and discharge planning in the hospital to improve future patient outcomes and reduce resource utilization. Whilst we did not assess the reasons for prolonged hospitalization in our study, high inappropriate use of antibiotics may have contributed to the delay in recovery [83,97,98].

Suggested ASPs in the hospital center on clinician education surrounding guideline-recommended antibiotics, as well as prospective audits coupled with feedback. These measures are some of the well-studied interventions that have shown a positive impact on improving guideline compliance within hospitals and subsequent appropriate antibiotic use, especially in low-resource settings, including among African countries [80,81,82,84,99,100,101]. Key areas for ASPs in HTH include the management of sepsis among neonates and children, which is associated with high mortality rates, appropriate antibiotic use to prevent surgical site infections, and educational programs and audits to reduce unnecessary prescribing of ceftriaxone alongside any irrational combinations [60,64,77,91,101,102,103]. Alongside this, universities in Ghana need to ensure that all healthcare students are well-versed in ASP principles, including increasing their knowledge and skills in the use of guideline-recommended antibiotics, as well as the WHO AWaRe system and guidance, by integrating this in their curricula and using teaching methods, including case-based learning and objective-structured clinical examination [60,84,100,101,104].

We are aware of a number of limitations with our study. Firstly, we only included one tertiary hospital for the reasons stated. The number of surveyed patients may also have been enhanced if the study duration had been prolonged. We also did not assess the impact of appropriate/inappropriate antibiotic use on outcome measures, including death, recovery, and discharge rates. Another observed limitation was the absence of microbiological data to support clinical decisions made; however, this is typically seen among these hospitals in Ghana with high patient co-payments for culture and sensitivity testing. We also did not have a comparator group, as we wanted to assess adherence to current guidelines in this vulnerable group as a starting point going forward. Finally, we did not assess issues of seasonality for the same reasons. Despite these limitations, we are confident that the study findings have provided baseline data for more rigorous studies and implementation research to improve antibiotic use among the pediatric population in this hospital and across the region.

## 4. Materials and Methods

### 4.1. Study Design

The study design was a prospective cross-sectional survey of the medical records of all pediatric patients (children under 12 years) admitted and treated with antibiotics at the Ho Teaching Hospital (HTH) over a three-month duration (January 2022 to March 2022) using the hospital’s electronic database.

The HTH was chosen for this study as it is a tertiary care facility located in the capital city of the Volta region, established in November 1998. It is currently the main referral facility in the Volta region [61]. The Volta region is one of Ghana’s sixteen administrative regions, with a 2021 population of 1,659,040, and Ho is its capital city [105]. The HTH provides services to approximately 194,000 outpatients and 12,500 inpatients annually. Alongside this, their gatekeeper system allows for admission and management of not only tertiary but also primary care cases. Consequently, the findings of antibiotic use from this facility may apply to lower (primary) and upper (secondary and tertiary) care facilities with a similar demography and resources in this country and other LMICs.

### 4.2. Study Site and Population

This study was conducted in the Pediatric Department of the HTH (comprising the Neonatal Intensive Care Unit (NICU), Babies Unit, and Children’s Unit) located in the Volta region of Ghana. The Pediatric Department admits sick children who are less than 1 month in the NICU, between 1 month to 6 months in the Babies Unit, and between 7 months and less than 12 years in the Children’s Unit. The Children’s Unit with 25 beds admits an average of 950 children annually, the baby unit with a 20-bed capacity admits an average of 700 babies, and the NICU with a 20-bed capacity admits an average of 450 neonates annually [61]. The department has pediatric consultants, specialists, residents, and medical officers providing various services to hospitalized and ambulatory care patients.

### 4.3. Inclusion and Exclusion Criteria

All pediatric patients (below the age of 12 years) admitted, diagnosed with an infectious disease, and managed with antibiotics were included in this study. All sick pediatric patients within the target age group who were diagnosed as having infectious diseases and managed with antibiotics in the ambulatory clinics of this hospital were excluded to concentrate just on inpatient prescribing.

### 4.4. Sample Size and Sampling Technique

No sampling was performed, as all hospitalized sick children who met the inclusion criteria were included in this study. However, the study sample was guided by an estimated size of 325 patients from an expected annual sample of participants using the Raosoft Inc. (Seattle, WA, USA) online calculator, assuming a 50% appropriateness of antibiotic prescription, an average monthly inpatient attendance at the pediatric department of 175, a 95% confidence interval, and a 5% margin of error.

### 4.5. Data Collection

Data were collected from the hospital’s Lightwave Health Information Management System (LHIMS) (an electronic medical record system) using an adapted data collection checklist from the Global Point Prevalence Survey Data Collection sheet [49,106,107]. The checklist included sociodemographic information, such as age, gender, and location of residence (rural, urban, or peri-urban), and clinical characteristics, including the treatment based on biomarkers, the duration of admission, culture and sensitivity testing, the number and type of antibiotics prescribed, the route of administration, and the diagnosis made.

Antibiotics were grouped according to their anatomical, therapeutic, and chemical (ATC) classification, as well as by their AWaRe classification [22,35,108]. AWaRe antibiotic groups were combined if a pediatric patient was given more than one antibiotic from different groups, e.g., Access and Watch antibiotics. The long-term goal of the UN GA is to increase the use of Access antibiotics to at least 70% utilization across sectors [43].

### 4.6. Antibiotic Appropriateness

Antibiotic appropriateness was assessed based on the right choice of antibiotics according to the antibiotic guidelines, as contained in the Seventh Edition of the Ghana Standard Treatment Guidelines (STG) for the management of infectious diseases in the pediatric population [92]. Compliance with current guidelines is increasingly seen as an important marker of the quality of antimicrobial prescribing [50,52,74,104,109,110], especially if no or limited CST testing is undertaken, as typically seen among hospitals in Ghana, with high co-payments for these tests [49,51,58,60].

### 4.7. Data Analysis

The checklist data were entered into a Microsoft Excel sheet before being exported to STATA version 14 (StrataCorp, College Station, TX, USA) for analysis. The appropriateness of prescribed antibiotics based on the choice for the diagnosed infectious disease was the primary study outcome measure.

Descriptive statistics were used to determine the mean age, the duration of admission, the duration of antibiotic use, and the proportions of each categorical variable. Diagnoses were combined where there were small patient numbers to help improve the interpretation of the findings. The chi-square test of independence was performed to assess any association between the outcome variable and the various independent variables. A logistic regression analysis was also performed using all statistically significant independent variables from the bivariate analysis (*p*-value < 0.05 at a 95% confidence interval).

### 4.8. Ethical Consideration

Ethical approval was secured from both the Research Ethics Committees of the University of Health and Allied Sciences (UHAS-REC A.7 [75] 22-23) and the Ho Teaching Hospital (HTH-REC (33) FC_2022). Personal identifiers of patients were excluded during the data collection to safeguard confidentiality. No informed consent was sought from the participants as there was no direct patient contact during the data collection period. The medical records of patients were directly extracted from the hospital electronic database.

## 5. Conclusions

This study found a high prevalence of antibiotic use in this hospitalized pediatric population. This included irrational broad-spectrum antibiotic combinations and third-generation cephalosporins, with ceftriaxone being the most commonly prescribed. These are concerns, as this combination potentially increases antibiotic resistance, highlighting the need for more responsible prescribing practices.

The hospital should prioritize continuous professional development on rational prescribing as part of ASPs. ASPs should concentrate on increasing adherence to evidence-based guidelines, including for surgical prophylaxis and neonatal sepsis through training and education on these guidelines and the WHO AWaRE classification. There is a need to instigate diagnostic AMS by improving microbiological infrastructure and resources to address concerns with currently high patient co-payments for CST to reduce empiric prescribing in this and other hospitals in Ghana, along with sustained surveillance and monitoring of antibiotic use and antimicrobial susceptibility data, all driven by strong leadership commitment.

## Figures and Tables

**Table 1 antibiotics-14-00779-t001:** Sociodemographic and clinical characteristics and their association with guideline compliance.

Variable	Total, n (%)	Compliance with the Guideline		*p*-Value
Age (years), mean ± SD	2.39 ± 2.99	Yes, n (%)	No, n (%)	
Duration of admission (days), mean ± SD	9.14 ± 10.73			
Duration of antibiotics (days), mean ± SD	3.03 ± 2.43			
Age (years) (n = 319)				0.393
0–2	219 (68.65)	99 (45.21)	120 (54.79)	
3–5	53 (16.61)	20 (37.74)	33 (62.26)	
6–8	26 (8.15)	10 (38.46)	16 (61.54)	
9–11	21 (6.58)	6 (28.57)	15 (71.43)	
Gender (n = 319)				0.591
Male	174 (54.55)	76 (43.68)	98 (56.32)	
Female	145 (45.45)	59 (40.69)	86 (59.31)	
Residence (n = 318)				0.836
Rural	149 (46.86)	62 (41.61)	87 (58.39)	
Urban	144 (45.28)	61 (42.36)	83 (57.64)	
Peri-urban	25 (7.86)	12 (48.00)	13 (52.00)	
Duration of admission (days) (n = 319)				0.569
1–5	153 (47.96)	69 (45.10)	84 (54.90)	
5–10	87 (27.27)	36 (41.38)	51 (58.62)	
>10	79 (24.76)	30 (37.97)	49 (62.03)	
Treatment based on biomarker (n = 319)				0.304
Yes	308 (96.55)	132 (42.86)	176 (57.14)	
No	11 (3.45)	3 (27.27)	8 (72.72)	
If yes, (n = 308)				0.262
WBC	128 (41.56)	49 (38.28)	79 (61.72)	
WBC + CRP	81 (26.30)	38 (46.91)	43 (53.09)	
WBC + urine R/E	58 (18.83)	25 (43.10)	33 (56.90)	
WBC + CRP + urine R/E	20 (6.49)	7 (35.00)	13 (65.00)	
Other	21 (6.82)	13 (61.90)	8 (38.10)	
Number of antibiotics (n = 319)				0.546
1	120 (37.62)	46 (38.33)	74 (61.67)	
2	181 (56.74)	83 (45.86)	98 (54.14)	
3	15 (4.71)	5 (33.33)	10 (66.67)	
4	3 (0.93)	1 (33.33)	2 (66.67)	
Antibiotic type (n = 319)				<0.001 *
Ceftriaxone	66 (20.69)	24 (36.36)	42 (63.64)	
Cefotaxime + flucloxacillin	51 (15.99)	0 (0.00)	51 (100.00)	
Ampicillin + gentamicin	48 (15.05)	48 (100.00)	0 (0.00)	
Amoxicillin + clavulanic acid	24 (7.52)	10 (41.67)	14 (58.33)	
Ceftriaxone + metronidazole	21 (6.58)	9 (42.86)	12 (57.14)	
Other ^	109 (34.17)	44 (40.37)	65 (59.63)	
WHO AWaRe * category (n = 317)				<0.001 *
Access	86 (27.13)	63 (73.26)	23 (26.74)	
Watch	95 (29.97)	40 (42.10)	55 (57.89)	
Access + Watch	136 (42.90)	32 (23.53)	104 (76.47)	
Route of administration (n = 319)				0.009 *
Intravenous	264 (82.76)	108 (40.91)	156 (59.09)	
Oral	15 (4.70)	6 (40.00)	9 (60.00)	
Topical	5 (1.57)	0 (0)	5 (100.00)	
Intravenous + oral	27 (8.46)	19 (70.37)	8 (29.63)	
Other	8 (2.51)	2 (25.00)	6 (75.00)	
Diagnosis (n = 319)				<0.001 *
Neonatal sepsis	77 (24.14)	33 (42.86)	44 (57.14)	
Neonatal jaundice due to sepsis	18 (5.64)	11 (61.11)	7 (38.89)	
Pneumonia	46 (14.42)	39 (84.78)	7 (15.22)	
Surgical wound prophylaxis	46 (14.42)	2 (4.35)	44 (95.65)	
Tonsillitis	17 (5.33)	11 (64.71)	6 (35.29)	
Other #	115 (36.05)	44 (38.26)	71 (61.74)	

NB: *p*-values with the asterisk (*) symbol are statistically significant; #—other diagnosis included bacterial gastroenteritis (n = 6; 1.88%), cellulitis (n = 6; 1.88%), acute chest syndrome (n = 5; 1.57%), vaso-occlusive crisis (n = 5; 1.57%), burns (n = 3; 0.94%), acute gastritis (n = 3; 0.94%), chronic osteomyelitis (n = 2; 0.63%), urinary tract infections (n = 1; 0.31%), and cord sepsis (n = 1; 0.31%); ^—other antibiotics included amoxicillin + clavulanic acid + azithromycin (n = 6; 1.88%), ciprofloxacin + metronidazole (n = 6; 1.88%), ciprofloxacin (n = 5; 1.57%), ciprofloxacin + clindamycin (n = 4; 1.25%), benzylpenicillin (n = 3; 0.94%), phenoxymethylpenicillin (n = 3; 0.94%), amoxicillin (n = 2; 0.63%), ceftriaxone + azithromycin + flucloxacillin (n = 2; 0.63%) ceftriaxone + clindamycin + metronidazole (n = 2; 0.63%), cefuroxime + clindamycin (n = 2, 0.63%), and metronidazole (n = 2; 0.63%); * AWaRe = Access, Watch, Reserve [22]; WBC = white blood count; CRP = C-reactive protein; R/E = routine examination.

**Table 2 antibiotics-14-00779-t002:** Logistic regression between independent variables.

Variable	aOR	95% CI	*p*-Value
*Antibiotic name*			
Amoxicillin + clavulanic acid (R)	1.00		
Ceftriaxone	0.12	0.02–0.95	0.044 *
Cefotaxime + flucloxacillin	1.00		
Ampicillin + gentamicin	1.00		
Ceftriaxone + metronidazole	0.38	0.05–3.09	0.368
Other	0.35	0.07–1.86	0.220
*WHO AWaRe category*			
Access (R)	1.00		
Watch	1.28	0.27–5.97	0.757
Access + Watch	4.86	0.12–1.95	0.309
*Route of administration*			
Intravenous (R)	1.00		
Oral	0.54	0.14–2.03	0.361
Topical	1.00		
Intravenous + oral	1.26	0.38–4.16	0.703
Other	0.38	0.05–2.61	0.322
*Diagnosis*			
Neonatal sepsis (R)	1.00		
Neonatal jaundice due to sepsis	1.00		
Pneumonia	15.38	3.30–71.62	<0.001 *
Surgical wound prophylaxis	0.07	0.01–0.42	0.004 *
Tonsillitis	1.00		
Other	1.80	0.50–6.53	0.371

NB: *p*-values with the asterisk (*) symbol are statistically significant; AWaRe = Access, Watch, Reserve [22]; R = reference variable.

## Data Availability

Additional data will be made available from the co-authors upon reasonable request.

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
