# Peer review of "Antibiotic Use in Pediatric Care in Ghana: A Call to Action for Stewardship in This Population"

_antibiotics, 2025, doi:10.3390/antibiotics14080779_

Round 1
Reviewer 1 Report (New Reviewer)
Comments and Suggestions for Authors
Strengths:
Relevant topic with high public health importance
Detailed analysis of prescribing patterns: e.g. AWaRe classification
Statistical analysis
Actionable conclusions: authors offer clear recommendations for antibiotic stewardship programs
Weaknesses:
Single-center study
Short observation period: seasonal variations should be considered
Retrospective design without clinical context: maybe antibiotics were needed in these children. A reasonable comparrison group is needed.
No microbiological data
Author Response
Strengths:
- Relevant topic with high public health importance
- Detailed analysis of prescribing patterns: e.g. AWaRe classification
- Statistical analysis
- Actionable conclusions: authors offer clear recommendations for antibiotic stewardship programs
Weaknesses:
- Single-centred study
- Short observation period: seasonal variations should be considered
- Retrospective design without clinical context: Maybe antibiotics were needed in these children. A reasonable comparison group is needed.
- No microbiological data
Author comments: Thank you for these comments. We have now updated the paper incorporating these limitations and including why no comparison data and why no microbiological data. We trust this is now OK.

Reviewer 2 Report (New Reviewer)
Comments and Suggestions for Authors
Authors have extracted relevant data which may be useful for awareness in prescribing appropriate antibiotics. A Few Tables can be expressed in graphical form for better understanding, if possible.
Authors need to mention what technology was used to identify the organisms isolated and what technology was used for AST in this particular hospital -was it conventional or automated?

Author Response
Comments and Suggestions for Authors
Authors have extracted relevant data which may be useful for awareness in prescribing appropriate antibiotics.
Author comments: Thank you – appreciated!
2. Few Tables can be expressed in graphical form for better understanding, if possible.
Author comment: Thank you for this comment. However, we believe based on our multiple publications in this area, coupled with working with multiple co-authors across many countries and continents in recent years, including health authority personnel in different countries, that presenting data in a tabular format is easier for readers, etc., to read, digest and act upon. We hope you now agree with us.
3. Authors need to mention what technology was used to identify the organisms isolated and what technology was used for AST in this particular hospital -was it conventional or automated?
Author comment: We did not present any microbiological data as none was available. This is typical for hospitals such as these in Ghana with high patient co-payments for culture and sensitivity testing – we have included more details regarding this (and accompanying references in the updated paper). We only included biomarker data that were used to support clinical decisions. We hope this has clarified the situation.

Reviewer 3 Report (New Reviewer)
Comments and Suggestions for Authors
- The content in lines 58–61 is unclear. Make it clearer and easier to understand.
- Which infections were most frequently treated with antibiotics in the mentioned population?
- The authors stress the importance of teaching students and healthcare professionals about AWaRe and prescribing in accordance with guidelines. What methods might be used to strengthen this understanding in Ghanaian nursing and medical curricula?
- According to guidelines, just 43.2% of prescriptions were found to be suitable. What obstacles could be causing prescribers to adhere so poorly?
- What particular difficulties are identified in ASP implementation in LMICs such as Ghana, and how may they be practically resolved within the framework of the current healthcare system?
- What measures do the authors recommend to regulate the usage of antibiotics by Ghanaian pediatric hospital patients?
- As stated, antibiotics are frequently used to treat pneumonia and newborn sepsis have the authors looked into the mortality rate from AMR in the population receiving treatment for these illnesses?
Author Response
Comments and Suggestions for Authors
1. The content in lines 58–61 is unclear. Make it clearer and easier to understand.
Author comments: Thank you for this comment. We have now clarified those statements in the above-mentioned lines to increase clarity and ease of reading. We hope it's okay now.
2. Which infections were most frequently treated with antibiotics in the mentioned population?
Author comments: The infections mentioned in the diagnosis subgroup were those that had antibiotics prescription (as that was part of the inclusion criteria). Therefore, the commonest paediatric infectious disease for which antibiotics were used was neonatal sepsis. This has been mentioned in the paper, and we hope this is now OK.
3. The authors stress the importance of teaching students and healthcare professionals about AWaRe and prescribing in accordance with guidelines. What methods might be used to strengthen this understanding in Ghanaian nursing and medical curricula?
Author comments: Thank you for this feedback. We have now added details on the teaching methods for healthcare students to strengthen their knowledge and confidence in ASP principles.
4. According to guidelines, just 43.2% of prescriptions were found to be suitable. What obstacles could be causing prescribers to adhere so poorly?
Author comments: Thank you for this feedback. We have now updated the paper with some possible obstacles to prescriber adherence to guidelines, though these were not assessed in our study. We therefore recommended that future studies should consider assessing their impact on guideline adherence, and hope this is now OK.
5. What particular difficulties are identified in ASP implementation in LMICs such as Ghana, and how may they be practically resolved within the framework of the current healthcare system?
Author comments: Thank you for this. We have mentioned some of these difficulties and solutions (e.g poor guideline compliance, low awareness of WHO AWaRE classification) already in the introduction and discussion sections of the paper. We have now added extra details on this at the introduction section, and hope this is now acceptable.
6. What measures do the authors recommend to regulate the usage of antibiotics by Ghanaian pediatric hospital patients?
Author comments: Thank you for this feedback. We have mentioned the need for continuous professional development on rational prescribing, improvement on microbiological infrastructure and resources to reduce empiric prescribing, improved surveillance and monitoring of antibiotic and antimicrobial susceptibility data, etc., in the updated Discussion and Conclusion sections of the paper, and hope this is now OK.
7. As stated, antibiotics are frequently used to treat pneumonia and newborn sepsis. Have the authors looked into the mortality rate from AMR in the population receiving treatment for these illnesses?
Author comments: Thank you for this. We have now added in some recent data on mortality rates for AMR among children in SE Asia and Africa, and the potential reasons for this – including considerably increasing use of Watch and Reserve antibiotics in Africa – that urgently needs addressing. However – there is currently no such data in Ghana – although AMR rates are increasing which is a concern. We hope this is now acceptable to you.

Reviewer 4 Report (New Reviewer)
Comments and Suggestions for Authors
The manuscript presents a valuable assessment of pediatric cases at a Ghanaian hospital to evaluate the appropriateness of antibiotic use according to WHO guidelines. The topic is of clear public health significance, particularly in the context of global antimicrobial stewardship.
I recommend that the authors acknowledge the limited duration of the analysis period as a study limitation and consider expanding it in future research to strengthen the generalizability of the findings.
Additionally, to enhance the impact and relevance of the study, I suggest including patient outcomes, such as whether patients were discharged, recovered, or died, and correlating these with the appropriateness of antibiotic use. This would significantly deepen the clinical insight provided by the study.
Minor Corrections & Suggestions:
-
Results Section (Lines 132–155):
Please reference Table 1 when describing the initial findings. -
Clearly specify the comparator for all p-values (e.g., compared to what group or variable?).
-
Lines 159–167 (NB Section): Organize the “NB” points in a clear and numbered or bulleted format to improve readability.
-
Line 170: Clarify what the number refers to, as it is not found in Table 1—please provide a reference or explanation.
-
Appropriateness Criteria: It would be helpful to clearly describe the method or criteria used to assess the appropriateness of antibiotic use, either in the Introduction or Methods section.
Author Response
Comments and Suggestions for Authors
1. The manuscript presents a valuable assessment of pediatric cases at a Ghanaian hospital to evaluate the appropriateness of antibiotic use according to WHO guidelines. The topic is of clear public health significance, particularly in the context of global antimicrobial stewardship.
Author comments: Thank you for this – appreciated!
2. I recommend that the authors acknowledge the limited duration of the analysis period as a study limitation and consider expanding it in future research to strengthen the generalizability of the findings.
Author comments: Thank you for this. We have acknowledged this limitation at the last paragraph of the discussion of the paper, and hope this is now OK.
3. Additionally, to enhance the impact and relevance of the study, I suggest including patient outcomes, such as whether patients were discharged, recovered, or died, and correlating these with the appropriateness of antibiotic use. This would significantly deepen the clinical insight provided by the study.
Author comments: Thank you this feedback. We agree that adding these data would have enhanced this paper. We, however, did not assess these impact (outcome) measures, and have mentioned this as a limitation of the study. We hope this is acceptable to you.
Minor Corrections & Suggestions:
4. Results Section (Lines 132–155): Please reference Table 1 when describing the initial findings.
Author comments: We are grateful for this comment. We have now inserted Table 1 in the initial finding presentation.
5. Clearly specify the comparator for all p-values (e.g., compared to what group or variable?).
Author comments: Thank you for this comment. We have added the comparator where p-values are mentioned in the results sections, and hope this is now OK.
6. Lines 159–167 (NB Section): Organize the “NB” points in a clear and numbered or bulleted format to improve readability.
Author comments: Thank you this comment. We have now properly bullet-formatted this as suggested, and hope this is now acceptable.
7. Line 170: Clarify what the number refers to, as it is not found in Table 1—please provide a reference or explanation.
Author comments: Please take note that these numbers are under the diagnosis (sub-variables) in Table 1. We have now added some details to improve the clarity of that statement, and hope this is now OK.
8. Appropriateness Criteria: It would be helpful to clearly describe the method or criteria used to assess the appropriateness of antibiotic use, either in the Introduction or Methods section.
Author comments: Thank you for this. We have captured the definition of appropriateness under the method section with a sub-heading (i.e., antibiotic appropriateness), and hope this is now acceptable.

This manuscript is a resubmission of an earlier submission. The following is a list of the peer review reports and author responses from that submission.
Round 1
Reviewer 1 Report
Comments and Suggestions for Authors
Please find the attachment.

Author Response
Dear Author,
I have read your manuscript with interest. The paper presents an important issue, namely the misuse of antibiotics in the hospital care system. Below, I present my major concerns. Minor concerns include the typo mistakes or the lack of legends under tables.
Author comments. Thank you for your comments. We hope we have adequately addressed these where we can, and made suitable comments where we have concerns. In addition – we have addressed typo mistakes where identified.
A) Section: Introduction
i) As you focus on children in Ghana, in the introduction, more information should be given regarding the healthcare system in Ghana, infections occurring in the studied population, limitations in access to medical care, prevalence of the disease, and population characteristics.
Author comments: Thank you for this comment. We have now added more information on infection occurrence, healthcare challenges among, and antibiotic use among the paediatric population in Ghana. We hope this is now okay.
ii) Not every reader is familiar with the AWARE program, so please explain it in a more detailed manner, especially by presenting antibiotics in each category.
Author comments: Thank you – we have now updated this, and hope this is now OK
iii) The book AWARE was released in 2022. Since the publication date, some new guidelines have been released or not integrated into the Aware book, eg, recommendations on the treatment of the C. difficile infections.
See: Stuart Johnson, Valéry Lavergne, Andrew M Skinner, Anne J Gonzales-Luna, Kevin W Garey, Ciaran P Kelly, Mark H Wilcox, Clinical Practice Guideline by the Infectious Diseases Society of America (IDSA) and Society for Healthcare Epidemiology of America (SHEA): 2021 Focused Update Guidelines on Management of Clostridioides difficile Infection in Adults, Clinical Infectious Diseases, Volume 73, Issue 5, 1 September 2021, Pages e1029–e1044, https://doi.org/10.1093/cid/ciab549.
Thus, despite the high value of the Aware book, it should be mentioned that up-to-date international and/or national guidelines and recommendations should be analyzed to provide treatment based on the latest guidelines.
Author comments: Thank you for this. We have added in this reference.
iv) L 78-80 “More recently, the WHO has launched the WHO AWaRe book, giving guidance on the management of 35 infections typically seen across sectors [33-35]”. I recommend changing it to “In 2022, the WHO has launched the WHO AWaRe book […]” and replacing the references 33-35 with the reference of the book - Suggested citation. The WHO AWaRe (Access, Watch, Reserve) antibiotic book. Geneva: World Health Organization; 2022.
Author comments: Thank you for this. We have added in this reference. However, we have kept some of the peer-reviewed references also giving insights as this is a paper for an academic journal – and in our considerable experience references from high-impact peer-reviewed Journals are generally more welcomed than internet references. We hope you agree.
vi) Moreover, the aim of your study should be more clearly presented – evaluation of the antibiotics (over)use in a children’s hospital in Ghana.
Author comments: Thank you for this – now updated.
B) Section: Results
As the material and methods are described at the end of the manuscript, at the beginning of the results section, I would recommend giving a brief description of the study period, population, and data collected and subjected for analysis.
i) Author comments: Thank you for this comment. We have now added as an introduction to the result section a brief description of the study period, population, and how data was collected. We must, however, mention that this arrangement of the different sections is in accordance with the Journal guidelines
ii) L 112: change the dot to a comma after “day.”
Author comments: Thank you – now altered
iii) L 115: change “+” to “and” or “with”
Author comments: Thank you – now altered
iv) L 120 and L 122-123 contain the same text
Author comments: Thank you – now altered
v) L 126: remove an extra parenthesis at the end of the sentence.
Author comments: Thank you – now altered
vi) L 129: insert “%” after 14.42.
Author comments: Thank you – now altered
vii) Table 1 in legend add explanation for: urine R/E, blood C/S, wound swab C/S.
Author Comments: Thank you for this. We have now added explanations to the legend under Table 1.
viii) The division into antibiotic types and classes is not clear. Consider subtracting this part of the table into a new table and merging types and classes.
Author Comments: Thank you for this comment. We agree that putting these two into the Table is confusing. Consequently, we have now removed the class of antibiotics, leaving just the type of antibiotics and their frequencies in the revised Table 1. We trust this is now OK.
ix) L 14-141 and Table 2 present the same data.
Author comments: Thank you – now updated. We trust this is now acceptable.
x) L 153: change “co-amoxiclav” to “amoxicillin with clavulanic acid”.
Author comments: Thank you – now altered.
xi) Heading of Table 3: change dash to dot.
Author comments: Thank you – now altered.
xii) Table 3 What does “(R)” after “Amoxicillin + Clavulanic Acid”, “(R)” after “Access”, “(R)” after “Intravenous” mean and“(R)” after “Neonatal Sepsis” mean?
Author Comments: We have added the meaning of the letter R (Reference variable) to the legend under Table 3, and hope this is now OK.
xiii) L 160 add dot at the end of the sentence. 3
Author Comments: We have now updated this. Thank you
xiv) The names of antibiotics should be written in lowercase.
Author comments: Thank you – now altered.
xv) The study would be more interesting and valuable if the information regarding the amount of antibiotics were provided, eg, present as a DDD (daily defined doses).
Author comment: Thank you for this comment. However, we beg to differ. Some of the co-authors have been involved with multiple DDD studies with antibiotics (as well as other medicines) especially in ambulatory care – although there are appreciable concerns with the DDDs for paediatric patients (one of the co-authors has been involved with such discussions). However, the focus of this paper was more on guideline compliance, i.e. the appropriateness of the antibiotics prescribed rather than their specific quantity based on DDDs. In addition, their AWaRe category building on the recent UN GA goals for Access antibiotics. Consequently, we believe we have used the optimal metrics in this paper and we hope that you now agree with us.
C) Section: Discussion
This section should be more focused on the discussion of the observed in the survey antibiotics use and its confrontation with references regarding the antibiotics consumption in hospitals in a studied region and other parts of the world, and less focused on antibiotics stewardship programs.
Author comments: Thank you for this – we have made some adjustments here and in the introduction. Given the identified concerns regarding the use of antibiotics in this paediatric hospital – which is a training hospital for HCPs who will subsequently treat children in ambulatory care on qualification – we believe it is vital that antibiotic utilisation patterns are improved in this important sector and these trainee HCPs are part of it before qualification. This naturally starts with assessing current prescribing patterns against current guidelines and carried on into ASPs. As a result, this learned behaviour, including improved adherence to robust guidelines, can be carried forward in their future practices in ambulatory care treating children. We hope you now agree with us.
D) Section: Materials and methods
i) Section 4.1 The study design should present what was investigated and how. The text in lines 260-266 is your speculation, not a study design. Rephrase or remove this part.
Author Comments: Thank you for this comment. We have removed the speculative statement here; however, included these comments in the Introduction and Discussion as a key reason for this research. We trust this is now OK.
ii) References 57, 103 in this section (line 258, and line 259) are unnecessary, remove them.
Author comment: Thank you – however we beg to differ as in the opinion of many reviewers we have encountered (with one of the co-authors having now over 550 publications in peer-reviewed papers to his name) they typically ask for references for key statements such as these. We hope you now agree with us.
iii) Add a more detailed description of the hospital – number of beds, type of wards, hospitalization rate, doctors' specialties, access to medical microbiology diagnostic laboratory, access to consultants – clinical pharmacologist, medical microbiologist.
Author Comments: Thank you for this comment. We have now added in more information, and trust this is now acceptable.
iv) Section 4.2 For what reason is reference 57 present in the section titled Study Site and Population? Remove it.
Author comment: Now removed the first time mentioned. We believe though it is appropriate to include the second Ref 57 (as above), and again trust that you now agree with this.
v) Section 4.5 Replace the references 46 and 48 present in the section titled Data Collection with reference to Global PPS - https://www.global-pps.com/documents/
Author comments: Thank you – now added in as well. However – we believe it is important in an academic paper to also include exemplars of its use and trust you agree with this. Having said this, we have removed ref 48 and replaced this with the findings from the Global PPS study. We hope you now agree.
vi) You can also consider adding the data collection templates as supplementary materials.
Author comments: Thank you – although again we beg to differ having now given the website address for those researchers who are interested as this avoids any copyright issues. We hope this is now OK.
vii) The text supported by reference 40 was cited in lines 84-85, and it is useless to repeat this sentence in this section.
Author comment: Thank you – however, we again beg to differ as we believe this is a key part of the paper given the UN target of at least 70% of antibiotic use in given sectors being Access antibiotics (as you are aware the previous WHO target was 60% Access antibiotics). Consequently, future PPS and other studies should now have this target in mind. We trust you now agree with us.
vii) Subheading 4.5 is doubled: “4.5 Data Collection” and “4.5 Antibiotic appropriateness”. Change subheading numeration.
Author comments: Thank you – now altered.
viii) L 305-306, remove this sentence. It has nothing to do with materials and methods, and it seems to be an unjustified self-citation! - references 47, 49, 108.
Author comments: Thank you for your comment. However, we again beg to differ. The initial research conducted by one of the co-authors in Namibia showed that adherence to current guidelines was seen as a much more appropriate marker of the quality of care than previous WHO INRUD criteria (Niaz Q, Godman B, Massele A, Campbell S, Kurdi A, Kagoya HR, et al. Validity of World Health Organisation prescribing indicators in Namibia's primary healthcare: findings and implications. Int J Qual Health Care. 2019;31:338-45). Since then, we have seen e.g. the Global PPS study of Versporten et al document adherence rates as a marker if quality with further examples in old ref 108. It is also likely (Ref 37) that we will increasingly see QIs being developed and implemented based on the AWaRe classification and book as the use of the AWaRe system to measure/ improve antibiotic utilisation grows (Ref 39). Consequently, we would like to keep this sentence as a key part of our paper. We trust you now agree.
ix) Section 4.7 Remove the reference 75 given in lines 320 and 347 of the Ethical Consideration section. Paragraph Informed Consent Statement 4
Author Comments: The 75 mentioned in the ethical consideration section is not a reference but rather part of the ethical clearance number given by the facility ERC. An explanation on why there was no need for informed consent for this PPS study has also now been added in.
x) Remove the citation from this paragraph. The reference 109 is cited only in this paragraph, which makes it another unjustified self-citation!
Author comments: Thank you – we included these references to justify why we did not ask for patient consent building on the previous comment as the Journal asks for an explanation if no patient consent was asked. We trust you now agree.
E) Conclusions
i) L 325 reword “in this Region in Ghana” and specify which region.
Author Comments: We have now reworded that statement. Thank you.
ii) L 327-329 repat findings presented in the Results section.
Author Comments: Thank you for this. The statement has now been paraphrased as suggested (lines 344-347).
F) Review summary
i) The manuscript presents important data on antibiotic use in children's hospitals in Ghana. The title does not reflect the scope of the study. The aim of the study is not clearly stated. Results are difficult to follow and are not explored deeply. Discussion mainly refers to antibiotic stewardship and barely discusses the findings of the study with references regarding the antibiotics consumption in hospitals in a studied region.
Author Comments: We are grateful for your thorough review. We have painstakingly addressed these comments and concerns where we can, as well as updated the Title. Consequently, we hope this update has taken care of the major issues you raised on the title, aim, results, method and discussion sections where we are able.
ii) I hope my concerns help you improve the manuscript.
Author comments: Thank you for your comments. We have tried to address these where we can and explained where we have difficulties or concerns with your comments. We hope you now agree with our changes.
Reviewer 2 Report
Comments and Suggestions for Authors
This is an epidemiological study on the use of antibiotics and its quality assessment in pediatric departments of a tertiary hospital in Ghana.
Several issues need to be addressed before the manuscript can be considered for publication.
a. The mean use of antibiotics in hospitalized patients is considerably shorter than the duration of their hospitalization; how is this explained? Were the infections for which antibiotics were prescribed considered community-acquired or healthcare-associated?
b. Why is ceftriaxone considered appropriate for the treatment of pneumonia?
c. How is culture-driven antibiotic treatment not considered appropriate? The authors need to elaborate further on this.
Author Response
This is an epidemiological study on the use of antibiotics and its quality assessment in pediatric departments of a tertiary hospital in Ghana.
Author comments: Thank you for this summary – appreciated.
Several issues need to be addressed before the manuscript can be considered for publication.
Author comments: Thank you for these comments. We hope we have adequately addressed these
a. The mean use of antibiotics in hospitalized patients is considerably shorter than the duration of their hospitalization; how is this explained? Were the infections for which antibiotics were prescribed considered community-acquired or healthcare-associated?
Author comments: Thank you for this. We did not categorize whether they were community-acquired or healthcare-associated infections in our results section. The possible reason for the longer duration of hospitalization compared to antibiotic use duration could be due to the late start of antibiotics plus the long period of observation after vital signs are stabilised. This may increase the risk of healthcare-associated infections. We have now amended this section, and hope that this is now OK.
b. Why is ceftriaxone considered appropriate for the treatment of pneumonia?
Author comments: Ceftriaxone was considered to be appropriate as the Ghanaian standard treatment guidelines recommend an option between ceftriaxone and amoxicillin + clavulanic acid for pneumonia in children (except neonates) and adults. This has been explained in the paper.
c. How is culture-driven antibiotic treatment not considered appropriate? The authors need to elaborate further on this.
Author comments: Thank you for this comment. We have removed that section from our results as the details were not fully accurate, and this does not affect the specific objectives of the study. We hope this is now OK.
Reviewer 3 Report
Comments and Suggestions for Authors
An interesting study of antibiotic use in paediatric inpatients in a hospital in Ghana. The results need expanding -Table 1 needs to be separated into several tables. Specifically antibiotic use needs to be stated in relation to diagnosis. Additionally, diagnoses and antibiotic treatment for the Other diagnoses need to be stated. This will allow readers to see where antibiotic prescribing is irrational and needs to be improved. Table 2 can be replaced by text. Logistic regression is unhelpful.
Discussion re what changes are planned in the hospital studied need to be added
Author Response
Comments and Suggestions for Authors
An interesting study of antibiotic use in paediatric inpatients in a hospital in Ghana.
Author comment: Thank you for this – appreciated.
1) The results need expanding -Table 1 needs to be separated into several tables.
Author comments: Thank you for this. We have removed some unnecessary sections from Table 1. We have also created Table 2 for assessment of guideline compliance as a separate dataset from the socio-demographic and clinical characteristics. However, we do not wish to break the revised Table 1 into additional Tables because we believe the revised Table 1 helps us to summarize our findings. We hope you now agree,
2) Specifically antibiotic use needs to be stated in relation to diagnosis. Additionally, diagnoses and antibiotic treatment for the Other diagnoses need to be stated. This will allow readers to see where antibiotic prescribing is irrational and needs to be improved.
Author comments: We have sections for diagnosis and antibiotic treatment. However, we merged the other diagnoses as they were many and had very small frequencies. This is because we believe that summarizing the Table in this way, and indicating what the other diagnoses entail in the notes at the bottom of the Table, helps with presenting and interpreting our findings. We hope this is OK with you.
3) Table 2 can be replaced by text. Logistic regression is unhelpful.
Author comments: Thank you for this comment. We though beg to disagree as we wanted to demonstrate what specific sub-variables predict the level of STG compliance. Presenting this in a narrative form without a table to demonstrate the other sub-variable could, we believe, make it difficult for readers to fully follow this. Overall, we believe we were able to show how the diagnosis of pneumonia increases compliance, but that of surgical prophylaxis reduces compliance. We hope this is now acceptable to you.
4) Discussion re what changes are planned in the hospital studied needs to be added
Author comments: Thank you for this. We have added in the discussion section the need for optimized care pathways and discharge planning to improve patient outcomes and resource utilization to deal with the problem of prolonged hospitalization. We have also added the need for Ghanaian universities to incorporate antimicrobial stewardship in the education of all healthcare professionals, emphasizing familiarity with guidelines, WHO AWaRe classification of antibiotics, etc. We trust this is now OK.
Round 2
Reviewer 1 Report
Comments and Suggestions for Authors
Dear Authors,
The revised version of your manuscript and the new title better represent the aim of the study and the scientific problem.
I have no further concerns.
Author Response
Dear Authors,
The revised version of your manuscript and the new title better represent the aim of the study and the scientific problem.
I have no further concerns.
Author comments: Thank you for your help – appreciated!

Reviewer 3 Report
Comments and Suggestions for Authors
Disappointing that the authors have failed to take on board the suggested changes. The paper in its current format is unsuitable for publication
Author Response
Comments and Suggestions for Authors
Disappointing that the authors have failed to take on board the suggested changes. The paper in its current format is unsuitable for publication
Author comments: Thank you – we are sorry to hear this. In view of this, we have now further revised the content of the paper and trust this is now acceptable.
Original comment: 1) The results need expanding -Table 1 needs to be separated into several tables.
Author comments:
a) Original: Thank you for this. We have removed some unnecessary sections from Table 1. We have also created Table 2 for assessment of guideline compliance as a separate dataset from the socio-demographic and clinical characteristics. However, we do not wish to break the revised Table 1 into additional Tables because we believe the revised Table 1 helps us to summarize our findings. We hope you now agree.
b) Update: We would still like to keep the revised Table 1 as it is (having now reduced some parts). We believe this helps the reader quickly assimilate key points together. We hope you agree with us.
Original comment 2) Specifically antibiotic use needs to be stated in relation to diagnosis. Additionally, diagnoses and antibiotic treatment for the Other diagnoses need to be stated. This will allow readers to see where antibiotic prescribing is irrational and needs to be improved.
Author comments:
a) Original: We have sections for diagnosis and antibiotic treatment. However, we merged the other diagnoses as they were many and had very small frequencies. This is because we believe that summarizing the Table in this way, and indicating what the other diagnoses entail in the notes at the bottom of the Table, helps with presenting and interpreting our findings. We hope this is OK with you.
b) Update: As seen – we have now added in details of a number of the other indications at the bottom of Tabkle 1. As also seen, the number of patients with these additional indicators are quite small (not all other indications have been included here – just some of these). In view of this, we did not include details of these various multiple indications in the main body of the paper (initially and subsequently) since we believed dividing up small numbers into guideline compliant or not would make for very big differences in % response rate if just one of these changed compliance levels, and create an undue focus anyway if large % numbers were involved. This is why we chose to combine all the diagnoses with small patient numbers together into one group as ‘other’. We still believed this is more meaningful to all key stakeholder groups going forward – especially as it helps to focus ASP activities where they are most needed. We have now altered the Methodology as well to make this point, and we hope that you now agree with us.
Original comment 3) Table 2 can be replaced by text. Logistic regression is unhelpful.
Author comments:
a) Original: Thank you for this comment. We though beg to disagree as we wanted to demonstrate what specific sub-variables predict the level of STG compliance. Presenting this in a narrative form without a table to demonstrate the other sub-variable could, we believe, make it difficult for readers to fully follow this. Overall, we believe we were able to show how the diagnosis of pneumonia increases compliance, but that of surgical prophylaxis reduces compliance. We hope this is now acceptable to you.
b) Update: We would still like to keep this Table as in our considerable experience when critiquing/ commenting on papers as well as educating students, etc., people more easily assimilate figures from a Table as opposed to the text. We have also added in further details as well to this section, and trust this is now acceptable.
Original comment 4) Discussion re what changes are planned in the hospital studied needs to be added
Author comments:
a) Original: Thank you for this. We have added in the discussion section the need for optimized care pathways and discharge planning to improve patient outcomes and resource utilization
b) Update: As seen – we have now added in further details about possible ASPs building on our considerable knowledge and experience in this area. We trust this is now acceptable.
Round 3
Reviewer 3 Report
Comments and Suggestions for Authors
The authors have still failed to address the points raised. As authors they have the right to submit their paper elsewhere, which is what I would recommend them to do